# Governing antibiotic resistance through One Health: Insights from the political and legal landscape in Senegal

Mouhamadou Moustapha Sow[1,2], Léo Delpy[3,4], Mamadou Ciss[2], Assane Gueye Fall[2], Nicolas Djighnoum Diouf[5], Jean-Hugues Caffin[4,6], Ndeye Mery Dia[1], Marion Bordier[2,4,6]*

1 Health Science and Research Laboratory, Gaston Berger University, Saint Louis, Senegal, 2 National Laboratory for Livestock and Veterinary research, Senegalese Institute of Agricultural Research, Dakar, Senegal, 3 UMR 8019, University of Lille, Villeneuve d'Ascq, France, 4 ASTRE, University of Montpellier, CIRAD, INRAE, Montpellier, France, 5 Laboratory of Biological, Agronomic, Food Sciences and Modelling of Complex Systems, Gaston Berger University, Saint Louis, Senegal, 6 CIRAD, UMR ASTRE, Dakar, Senegal

* marion.bordier@cirad.fr

## Abstract

Antimicrobial resistance (AMR), especially antibiotic resistance (ABR), could cause up to 10 million deaths per year by 2050. While ABR is a natural phenomenon, it is exacerbated by antibiotic misuse, poor sanitation, insufficient health infrastructure, weak biosecurity, and environmental contamination. The cross-transmission of resistant bacteria between humans, animal and the environment requires a multi-sectoral response inspired by the One Health concept. With the support of international partners and in line with global policies, Senegal has established a national One Health platform and legal measures to combat ABR. Our study proposes to examine the One Health governance capacities for managing ABR in Senegal, and to identify ways of improvement from a One Health perspective. A qualitative approach was adopted, combining the review of 64 legal instruments and 26 semi-structured interviews conducted between January and August 2023 with actors from governmental, private and non-governmental institutions. The analysis focused on governance structure, legal instrument development and implementation, and ABR surveillance programmes. The thematic analysis of the interview contents highlighted factors influencing the One Health governance capacities in Senegal. One Health governance has been institutionalized in Senegal since 2017 with the creation of the national One Health platform housed at the Prime Minister's Office. However, its effectiveness is limited by weak formalization, inadequate resources, and reliance on external funding. Outdated and poorly enforced legal frameworks hinder ABR management, while rural areas lack sufficient laboratory infrastructure, leading to inappropriate antibiotic use and weak surveillance systems. Power imbalances among stakeholders impede collective action and data-sharing. Despite progress toward international standards, systemic barriers persist. Strengthening the legal framework,

**Data availability statement:** The dataset that supports the literature analysis conducted during this study is available at: https://doi.org/10.18167/DVN1/DA4KSE. Interview transcripts are not publicly available due to ethical and confidentiality constraints.

**Funding:** This study is part of the Designing One Health Governance for Antimicrobial Stewardship Interventions (DESIGN) project, funded through the Joint Programming Initiative on Antimicrobial Resistance (JPI-AMR). MMS, MC, AGF, NDD and NMD were supported by the Swedish International Development Cooperation Agency (SIDA) [Grant #274846]; MB and LD by the French National Research Agency (Agence Nationale de la Recherche, or ANR) [Grant # ANR-21-AAMR-0002-03]; and MB, LD, and JHC by the French Agricultural Research Centre for International Development (Centre de coopération internationale en recherche agronomique pour le développement, or CIRAD). The funders had no role in study design, data collection and analysis, decision to publish, or preparation of the manuscript.

**Competing interests:** The authors have declared that no competing interests exist.

improving healthcare infrastructure, and enhancing the One Health platform's status and funding are critical. Institutionalization must connect global orientations with local realities to avoid inequities. Sustainable ABR management requires inclusive, context-sensitive policies, robust domestic investment, and equitable collaboration among stakeholders across the sectors of human, animal, and environmental health.

## Introduction

Antimicrobial resistance (AMR) has become one of the greatest health issues of our time [1]. In 2015, initial estimates of the health impact of AMR pointed to 10 million deaths per year by 2050, with an estimated impact on global GDP of $1 trillion per year [2]. Antibiotic resistance (ABR) plays a key role in this major health problem. A recent study in 2022 directly attributes 1.27 million human deaths to bacterial resistance in 2019, with sub-Saharan Africa (SSA) accounting for the majority [3]. ABR thus could compromise efforts to achieve sustainable development goals in low- and middle-income countries (LMICs) [4,5].

ABR occurs when bacteria become resistant to antibiotics that were previously effective against them. It is the result of natural selection, exacerbated by other factors such as antibiotic overuse and misuse, and insufficient access to drinking water and sanitation [6,7]. Resistant bacteria are found in the human, animal, and environmental sectors and can spread easily through direct contact, the food chain, and environmental pollution by effluents from livestock farms, hospitals and pharmaceutical facilities, and waste and sewage treatment plants [8–12]. Resistance is further amplified as bacteria exchange genetic material in shared ecosystems. Additionally, resistant bacteria and antibiotic residue can spread across geographical areas via trade and human travel [13,14]. Consequently, managing ABR requires a coordinated, multi-scale and multi-sectoral approach, in line with the One Health concept [15–17]. One Health is defined as an integrated, unifying approach that aims to sustainably balance and optimize the health of people, animals and ecosystems. This concept recognizes the links between and interdependence of the health of humans, domestic and wild animals, plants, and the wider environment (including ecosystems) [18].

In response to the ABR crisis, the international community has committed to implementing effective and systemic policy measures to combat ABR, including the Global Action Plan on AMR, adopted by the World Health Assembly and issued by the World Health Organization (WHO) in 2015 [19]. Additionally, sectoral strategies have been issued by the Food and Agriculture Organization (FAO) and the World Organisation for Animal Health (WOAH) for the food and the animal sectors, respectively [20,21]. These international plans provide guidance for countries to develop policy instruments and interventions, adapted to their national priorities and context, and promote effective multi-sectoral governance through the establishment of new politico-institutional arrangements to coordinate actions to combat ABR, in a One Health approach [22,23].

In line with these international recommendations, Senegal has adapted its existing politico-institutional framework to better manage the ABR issue, through the establishment of a multi-sectoral institutional mechanism to coordinate actions of stakeholders involved in ABR management and the development of inter-sectoral legal instruments [24,25].

In this context, this study aims to examine the capacity of the Senegalese politico-institutional framework to manage the ABR issue from a One Health perspective. Based on a review of the legal instruments related to ABR management and on interviews with key stakeholders, we describe the current political and institutional landscape in relation to ABR management, and identify determinants that could act as levers or barriers to effective One Health governance of ABR, as well as ways to improve this governance.

## Materials and methods

We applied a qualitative approach, combining a document review and semi-structured interviews with stakeholders to: (i) describe the existing politico-institutional framework for ABR management in Senegal, i.e., institutions involved with ABR management and the related legal instruments they have formulated and/or are implementing; and (ii) identify drivers that may hinder or favour One Health governance of ABR management in Senegal, and describe potential ways of improving towards a more effective One Health governance. The two methodological components are complementary and mutually reinforcing. The literature review allowed the description of the legal framework of One Health governance and the enrichment of our initial selection of interviewees. The interviews allowed the analysis of stakeholders' perceptions of the operationalization of this governance and its effectiveness, as well as the identification of new legal instruments related to One Health governance.

By ABR management, we refer to interventions that contribute to the prevention of the emergence and spread of ABR, through: (i) the reduction of the demand for antibiotics, including improving healthcare systems, biosecurity and hygiene practices; (ii) the judicious use of antibiotics, including reducing antibiotic misuse, controlling access to antibiotics, and improving antibiotic quality; (iii) the surveillance of antibiotic use, resistance and residues.

Governance can be defined as the processes by which governmental and non-governmental stakeholders within civil society and the private sector exercise their power and authority and use their resources to influence, develop, manage and implement policies at local, national and global levels [26]. For this study, we have adapted this definition to include the One Health dimension applied to ABR; by One Health governance of ABR, we refer to the process by which stakeholders collaborate and coordinate their collective action to develop and implement policies to manage ABR, sometimes underpinned by an institutional and legal framework, and facilitated by effective management, as well as transparent mechanisms and relationships.

### Ethics statement

This study was approved and authorized by the Senegalese National Ethics Committee for Health Research (*Comité National d'Ethique pour la Recherche en Santé*, or CNERS) in January 2023 (N°019MSAM/CNERS/SP). Before the start of each interview, participants were informed of this approval in addition to detailed information about the study objectives. In addition, informed consent was requested and anonymity in data processing was guaranteed in the case of each participant.

### Data collection

The data collection was carried out between January and August 2023 using two sources: a literature review, and interviews with key stakeholders involved in ABR management in Senegal.

Firstly, a review of the grey literature was conducted to compile a description of the politico-institutional framework for ABR management at regional and national levels, through a characterization of the stakeholders and of the legal instruments they have developed and/or are applying (policy, regulation, guidelines) to manage ABR. Documents were retrieved

by searching the website of the Senegalese government's Official Journal (*Journal Officiel,* https://www.vie-publique.sn/journal-officiel-senegal), and the FAO legal database FAOLex (https://www.fao.org/faolex). In addition, we consulted the official websites of organizations involved in ABR governance at continental and regional levels (the Africa Centre for Disease Control and Prevention of the African Union, the Regional Animal Health Centre, and the West African Health Organization of the Economic Community of West African States). Additional documents were also retrieved from scientific articles related to ABR management in Senegal [24] and in the West African Health Organization of the Economic Community of West African States [27], or suggested by participants to the interview. In total, 64 documents were selected for analysis.

In addition, we conducted semi-structured interviews with key stakeholders involved in ABR management to enrich our understanding of the existing politico-institutional framework and to collect their perspectives about One Health governance of ABR in Senegal. The recruitment of participants was conducted from 26 January to 30 September 2023 in two steps, as described by Laumann *et al.* (1983) [28]. First, we established an initial list of participants corresponding to the members of the thematic working group for AMR of the national One Health platform (67 people, each representing one institution). Then, we asked three experts in the field of ABR management in Senegal to revise the list. We ended up with 28 target participant institutions, which was extended to 29 during data collection as another key institution was mentioned during an interview. Official invitations were sent to the heads of each of the 29 institutions, who were requested to designate their most knowledgeable representative(s) with regard to the theme at hand. Semi-structured interviews were carried out using an interview guide consisting of six parts, all related to ABR management in Senegal: (1) the respondent's characteristics (professional background, role and missions in his/her institution); (2) the role of his/her institution in policy development and implementation; (3) the description of the national policy landscape; (4) the public policy development process (participation and modalities); (5) the implementation and enforcement of public policy; (6) the surveillance of antibiotic use and ABR; and (7) the coordination mechanisms for multi-sectoral governance (existing and potential future mechanisms). In total, we interviewed 37 people across 26 interviews (21 individuals interviews and five collective interviews), 92 percent of which were recorded. Interviews lasted between 60 and 90 minutes. They were conducted 21 interviews on face-to-face, the remaining (five) ones used a video conferencing platform. The whole study (data collection, management, and analysis) was conducted in French. All interviews were manually transcribed, with repeated listening and proofreading to ensure quality. Unrecorded interviews were faithfully reconstructed on the basis of written notes. The participants worked in the animal health sector (11), the agriculture and food sector (11), the human health sector (eight), and the environment sector (three). Four participants operated in a multi-sectoral organization. Participants belonged to: governmental authorities (12 people from seven institutions); scientific and technical institutions (14 people from 11 institutions); regional and international organizations (nine people from six institutions, three at regional level and three at international level); and the private sector (two informants from two institutions) (Table 1).

## Data analysis

Data analysis was carried out in two phases.

Firstly, we conducted a qualitative descriptive analysis of the contents of the documents retrieved during the literature review, and of other documents mentioned during the interviews, in order to characterize the politico-institutional framework, namely its composition of: (i) the institutions involved in ABR governance (sector, mission, institutional level); (ii) the legal instruments related to ABR (nature, implementation, intervention domain); and (iii) the surveillance programmes for antibiotic use, resistance and residues (regulatory status, objective, strategy). The attributes used to describe institutions and instruments were primarily retrieved from the literature [29] and refined during the analysis of the document and interview contents.

Secondly, a thematic analysis was carried out on the transcribed discourses of the informants, using the Nvivo software (version14) in order to: (i) deepen the characterization of the politico-institutional framework obtained with the document

**Table 1. Characteristics of the participants and of the interviews.**

| Sector | Activity domain | Number of institutions invited | Number of institutions interviewed | Number of experts interviewed in person | Number of experts interviewed online |
|---|---|---|---|---|---|
| Animal health | Governmental authorities | 2 | 2 | 4 | – |
| | Scientific and technical institutions | 1 | 1 | – | 1 |
| | Regional and international organizations | 3 | 3 | 1 | 3 |
| | Private sector | 2 | 2 | 2 | – |
| Agriculture and food | Governmental authorities | 1 | 1 | 1 | – |
| | Scientific and technical institutions | 5 | 5 | 8 | – |
| | Regional and international organizations | 1 | 1 | – | 2 |
| Human health | Governmental authorities | 1 | 0 | – | – |
| | Scientific and technical institutions | 6 | 5 | 3 | 2 |
| | Regional and international organizations | 2 | 2 | 1 | 2 |
| | Private sector | 1 | 0 | – | – |
| Environment | Governmental authorities | 3 | 3 | 3 | – |
| Multi-sectoral | Governmental authorities | 1 | 1 | 4 | – |
| Total | | 29 | 26 | 27 | 10 |

analysis; and (ii) identify the One Health capacities for ABR governance in Senegal and orientations for its improvement. The thematic analysis to investigate the One Health governance capacities used an inductive approach, following the process described by Braun and Clarke (2006): (i) familiarization with the data; (ii) generation of initial codes; (iii) generation of initial themes; (iv) review of the themes; (v) definition and naming of the themes; and (vi) writing up [30]. We also mobilized the analytical approach of Ritchie *et al*. (2002) to identify barriers to and determinants of effective One Health governance for ABR management and to formulate theme names [31]. The themes were refined following various discussions between members of the research team.

## Results

Following the analysis of the 64 documents retrieved and of the discourse of the 26 institutions interviewed, we analysed the historical development and the structure of the politico-institutional framework of ABR management in Senegal, its capacities to manage ABR using a One Health approach, and the necessary changes to move towards a more effective One Health governance of ABR in Senegal.

### The politico-institutional framework for antibiotic resistance management in Senegal

Since the early 2000s, Senegal has been developing a politico-institutional framework to support the management of ABR, both at sectoral and multi-sectoral levels, in line with international standards (Fig 1).

The National Programme to Combat Nosocomial Infections (*Programme National de Lutte contre les Infections Nosocomiales*, or PRONALIN) was a 10-year strategic plan (2005–2015) launched by the Ministry of Public Health in 2004, focusing on infections caused by resistant bacteria. It led to the establishment of committees to combat healthcare-associated infections in all hospitals and health centres. In 2012, under the PRONALIN framework, a department in charge of laboratories was established by the Ministry of Public Health to coordinate surveillance activities conducted by laboratories operating in the human, animal, and trade sectors. In 2017, this department led the development of the first national action plan under a One Health approach for a five-year period (2018–2022), in line with the Global Action Plan to combat AMR issued in 2015 by the WHO. The objective was to provide an effective response, through a One Health approach, to the threat of AMR, including ABR. This policy, developed by technical departments of ministries in charge

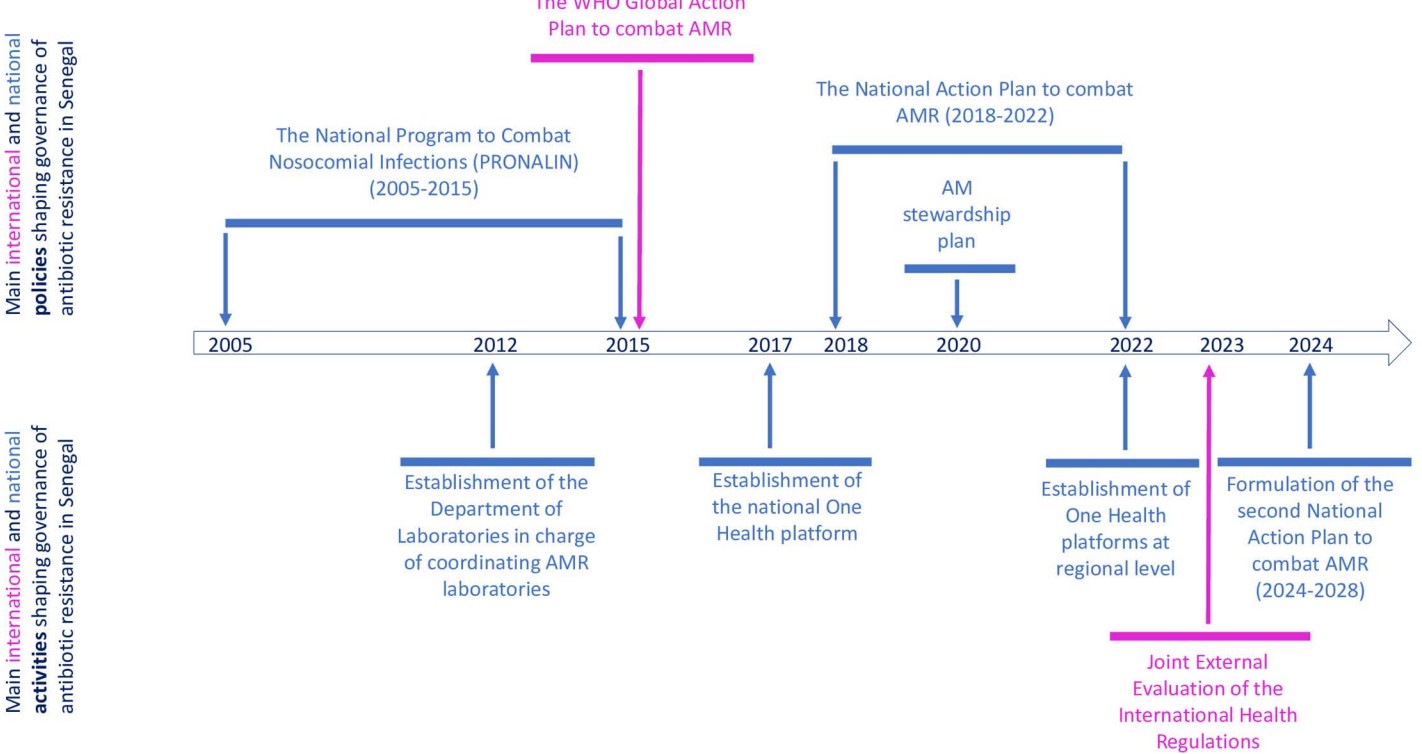

**Fig 1. Milestones of the development of the politico-institutional landscape for the management of antibiotic resistance in Senegal.** AMR = antimicrobial resistance.

of human and animal health, never obtained an official endorsement by the ministries, which compromised its effective implementation. However, legal instruments were issued to support the implementation of some of the measures included in the national action plan.

With the support of international partners (e.g., WHO and FAO), in 2017 Senegal established a national multisectoral platform named the National High Council for Global Health Security (*Haut Conseil national de la sécurité sanitaire mondiale*), under the prime minister. This national One Health platform is in charge of defining the strategic orientations of the One Health global health security programme [32] in compliance with the International Health Regulations (IHR) [33]. It includes a permanent secretary and 14 thematic working groups, aligned with the main domains addressed by the IHR, one of which is dedicated to AMR. These groups bring together representatives of the different ministries, academia, technical and financial partners (TFPs), decentralized authorities, professional organizations and civil society. The national One Health platform is currently in a process of decentralization, with the establishment of similar institutional arrangements at the regional level (the One Health Regional Committees), under the authority of the governors. So far, these regional platforms are barely operational, except in the region of Saint Louis where a regional One Health action plan was developed in 2024. In 2020, the AMR thematic working group of the national One Health Platform drafted an antimicrobial stewardship plan, under the impetus of the Medicines, Technologies, and Pharmaceutical Services project financed by the United States Agency for International Development (USAID), which at the time of writing has been barely operationalized. In October 2022, this same group took the lead to formulate the second national action plan to combat AMR, which was finalized in March 2024. Following the orientations of the IHR, the national action plan sets key priorities for combating AMR. The development of this second plan also allowed the country to comply with one

criterion assessed by the Joint External Evaluation (JEE), the tool developed by the WHO to evaluate the level of compliance of the countries regarding the IHR provisions.

The Senegalese politico-institutional framework is also embedded in the continental and regional one. The African Union has issued several instruments to harmonize AMR management across its member countries, such as an action plan [34], a communication plan [35], and an antimicrobial surveillance framework [36]. The Economic Community of West African States has established a Regional One Health platform, which is currently barely functional, that brings together the Regional Animal Health Centre and the West African Health Organization, and plays a role in the harmonization of national policies and regulations at the regional scale. The Regional Drug Committee of the West African Economic and Monetary Union has issued regulations related to marketing authorization for medicinal products, including antibiotics, for animal health (2006) and human health (2020), allowing the monitoring of products' circulation and ensuring their quality in the region.

A representation of the Senegalese institutional framework in the international and regional context is presented in Fig 2.

In addition to the national action plans cited above, 40 other legal instruments have also been developed over the years. They consist of 27 binding legal instruments, namely laws (14), decrees (eight), orders (two), circulars (two), and standard (one), and 13 non-binding instruments, namely plans (two) and technical guidelines (11). The binding instruments target the prevention of ABR (14), the surveillance of antibiotic use and resistance (10), or the use of antibiotics (three). The non-binding instruments target the prevention of ABR (six), the surveillance of antibiotic use and resistance (five), or the use of antibiotics (two). Overall, the largest number of these legal instruments are found in the human (18) and the livestock sectors (nine) followed by the environmental sector (eight), at multi-sectoral level (trois), and in the agriculture and food sector (two). Regarding the prevention of ABR, emphasis is put on the implementation of biosecurity measures and good hygiene practices. The instruments for antibiotic use focus on regulating their access, providing guidelines for their judicious use, or controlling their promotion. In addition, the country counts eight public surveillance

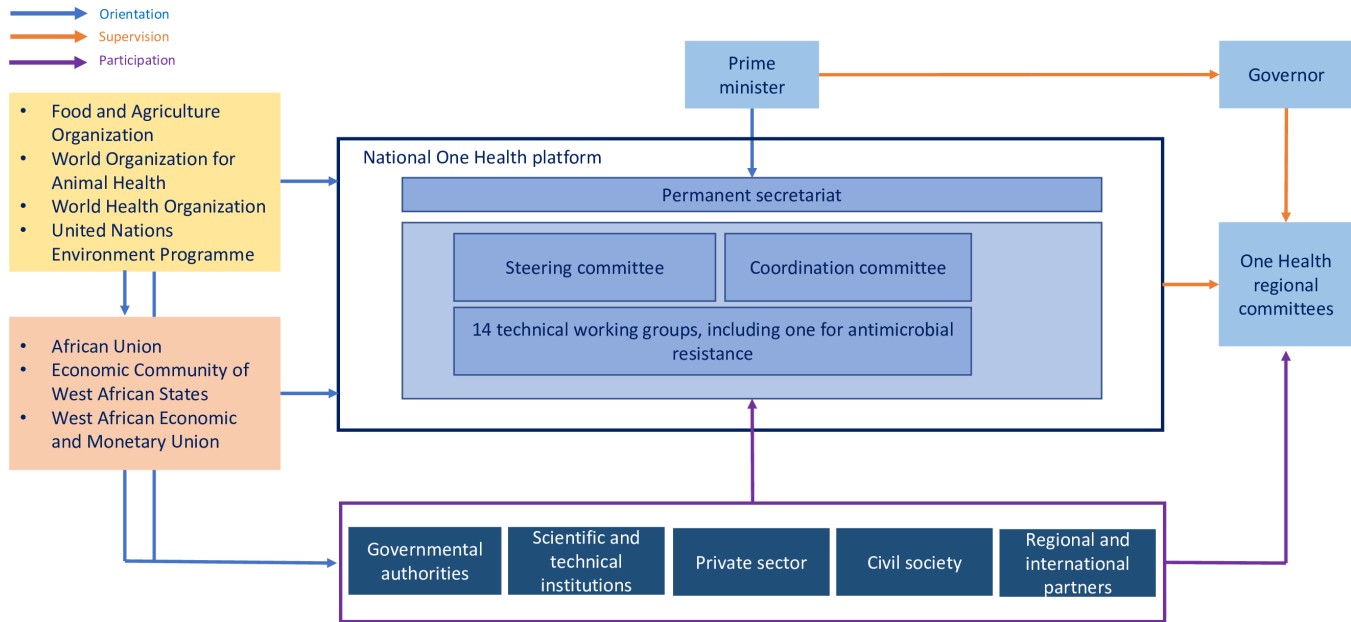

**Fig 2. The Senegalese multi-sectoral institutional framework involved in the management of antibiotic resistance and embedded in the global landscape.**

programmes, of which two are for ABR, three for antibiotic use (ABU), two for antibiotic residues, and one for antibiotic quality. These surveillance programmes are operating with very little interconnectedness but efforts are ongoing to integrate part of the data they are generating.

Institutions and their main legal instruments are described in S1 Table, and the surveillance programmes in S2 Table.

**The One Health governance capacities for antibiotic resistance in Senegal, limits and ways of improvement**

The analysis of the interview contents with regard to the existing politico-institutional framework allowed us to describe the national capacity in managing ABR from a One Health perspective, and to identify opportunities to move towards a more effective One Health governance of ABR. We identified five determinants that influence the One Health governance capacities for ABR management in Senegal.

**The existence of a multi-sectoral institutional arrangement for the governance of antibiotic resistance.** The national One Health platform has contributed significantly to the formulation of multi-sectoral legal instruments related to ABR, including the second national action plan, as illustrated by the following comment from Respondent 17: "*Since the platform was established, we have transferred to it the tasks of intersectoral development* [of legal instruments]*, as they are strategic, multisectoral and multidisciplinary documents, and therefore they must be truly inclusive and participatory*".

However, there are two main obstacles to the platform's effective operation. First, its roles and missions are insufficiently formalized, as indicated by Respondent 5: "*The platform is not very well equipped at the institutional and political level. It still seems to be legally weak.*". This hampers its ability to mobilize the national stakeholders towards the development of a common vision for the effective management of ABR. Additionally, the platform has insufficient domestic budget and human resources to carry out its missions and mainly relies on the financial support of international partners, as described by two respondents: "*If one does not have financial autonomy, it is very difficult to operate. For One Health coordination, you need a team, you need experts in the field.*" (Respondent 23) "*As ever, money talks. If you have a very nice car and you don't have the gas, it won't move.*" (Respondent 24)

If the national One Health capacities for ABR management are to be improved, the platform's legitimacy in relation to sectoral institutions needs to be strengthened, and a budget commensurate with its mandate need to be allocated, as suggested by Respondent 1: "*Strengthening the current platform should be enough. It should be enough if they just strengthen the legal framework and allocate sufficient budget.*"

**The influence of the technical and financial partners for the management of antibiotic resistance.** The TFPs are very active in the field of ABR management in Senegal. Numerous projects are financed and/or implemented by them in various fields, such as communication about the judicious use of antibiotics (Breakthrough Action project financed by the United States), surveillance of ABR and antibiotic usage (financed by the Fleming Fund), or specific interventions for the judicious use of antibiotics (financed by the Antimicrobial Resistance Multi-Partner Trust Fund). However, most programmes initiated and financed by international aid have difficulty surviving when external funding ceases, due to a lack of ownership by the government, which does not take over with domestic resources to maintain them. For instance, a surveillance programme for resistant bacteria in the agri-food sector has been established with FAO support, but national actors do not receive appropriate domestic budgets to keep it running. They are still relying on one-off external fundings to conduct surveillance activities, which need to be carried out in a timely manner, as indicated by a representative: "*Currently, we do not do antibiograms because we don't have any kits. I have made a new request to the FAO. We are waiting and I think they will say yes. They have launched the tender, I am waiting to see*" (Respondent 12). In addition, when formulating policies, national actors tend to prioritize interventions that will attract the attention of donors or that comply with their priorities and agendas: "*When policy is formulated, if activities are proposed that are not considered by, or are not in the area of interest of the partners, then [those activities] will not come to much. The key is to be supported by technical and financial partners*" (Respondent 3). Finally, it was recognized that local actors were often insufficiently consulted during policy development workshops organized by TFPs and that policy solutions did not reflect the realities on

the ground, as underlined by Respondent 13: *"There is a gap between what we local actors need and what is in the policy documents. During the workshops* [for the formulation of ABR policies], *we have bureaucrats who will think about and want to implement strategies in the field without involving those working on the ground in these discussions.".*

**The legal framework shaping the governance of antibiotic resistance.** The analysis of the legal framework shows that Senegal has a fairly comprehensive regulatory framework for managing ABR. However, this framework faces major problems. First, some provisions of the prevailing legal instruments are outdated regarding the rapidly changing context of ABR emergence and spread, as related by Respondent 3: *"There is a weakness in the regulation compared to these provisions* [on the storage and use of antibiotics]. *Because this* [ABR] *is an emerging phenomenon that has not been sufficiently taken into account."* This shortcoming is particularly alarming for the regulation on the use of antibiotics, as antibiotics can in practice often be accessed freely even by people who are not entitled to them: *"The regulations are not precise and there is no specification of standards so people deliver drugs without prescription and practice 'Kalashnikov treatments'* [using critically important antibiotics to treat mild infections]*)"* (Respondent 8). Furthermore, although the institutions responsible for implementing legal instruments are well-defined and have the necessary powers, they suffer from a lack of human and financial resources to carry out their supervisory and control tasks*, as described by Respondent 21: "For veterinary drug inspection, there are only three specialized inspectors at the national scale. It is very insufficient and, even worse, the logistical and financial means are insufficient."* Moreover, the application and enforcement of the law is hampered by passive corruption that is favoured by friendly or family relations: *"Unfortunately, we lack seriousness in the application of these texts. We have a term for that, called the* massla: *If I'm friends with someone and he is doing wrong things, I will consider it is ok, and this culture is holding us back* [in the application of laws]*"* (Respondent 22).

Moving towards more effective ABR management requires the updating and enforcement of the prevailing legal frameworks in different sectors, as suggested by Respondent 9: *"We need to update certain regulatory provisions among the ones existing and ensure that they are applied. This will enable us to combat ABR effectively."*

**The capacity of the healthcare system.** The literature review found that Senegal's healthcare system has improved significantly, particularly following the various health crises that have affected the country (Ebola, COVID-19), and with a particular focus on equity and inclusiveness. More specifically, committees in charge of combatting nosocomial infections have been established in hospitals to improve infection prevention and hygiene in healthcare environments, with a focus on ABR [37,38]. However, the healthcare system is not appropriately equipped at national scale to contribute effectively to the fight against ABR. In some remote areas, no bacteriology laboratories are available in either the human or the animal sectors. Where laboratory facilities exist, they face shortages of reagents for lab testing. As a result, few diagnostic results are available to guide the judicious prescription of antibiotics; practitioners sometimes use antibiotics when not needed or the wrong class of antibiotics for a particular bacterial strain (for instance critical antibiotics for benign infections). This is reported by Respondent 15: *"Not all hospital facilities are well equipped with laboratories that can carry out treatments based on antibiograms. Consequently, treatment is probabilistic."* This lack of diagnostic capacities also impacts the quality of the surveillance data, as underlined by Respondent 14: *"It's clear that there are bacteria in remote areas, and there are resistances there. But if we don't have a bacteriology laboratory capable of carrying out sensitivity tests, this information escapes us."*

To be able to properly manage ABR, Senegal must strengthen its surveillance capacities to appropriately inform the policy-making process. Among other things, this requires building capacity to conduct laboratory testing and to produce data at national scale, as suggested by Respondent 8: *"A staff training programme would enable laboratories in the country to have the capacity to carry out sensitivity tests."* This would also require the official appointment of an institution capable of collating all the data produced in the country. For the human sector, the national public health laboratory could play this role, as described in the case of Nigeria by Respondent 23: *"In Nigeria, there is only one structure that manages everything. The CDC* [Centre for Disease Control and Prevention] *manages all surveillance, the laboratory, the response and the fight against ABR, and all the information converges there."*

**The power balance among stakeholders.**  In recent years, with the establishment of the One Health Platform, national stakeholders have developed substantial experience in both formal and informal collaboration for managing health issues through a One Health perspective. This is illustrated by Respondent 13's comment: "*Within the thematic groups* [of the One Health Platform]*, we work together during workshops. We have developed cross-sectoral collaboration for designing strategy and also in the field.*" However, this collaboration runs into power issues between stakeholders. First, some professions operating in the human health sector, and to a lesser extent those working in the animal health sector, consider that they have a monopoly when it comes to public health, as observed by Respondent 22: "*During these workshops* [for the formulation of the second national action plan]*, each corporation pulls on its side; that is to say, if we talk about One Health, for example, the* [medical] *doctors always want to be the profession who leads. We forget about the environmentalist, we forget about the veterinarian, we forget about the pharmacist.*" They tend to overlook the points of view and perspectives of other professions, which are usually less directly involved in public health, and this hampers the design of collective solutions for an effective management of ABR. Respondent 12 recognized that, during the drafting of the second national action plan, "*a certain censorship is to be seen, which can effectively block the whole group's progress.*" Power issues were also described in the field of ABR surveillance, when it comes to sharing data across professions and sectors in an effort to provide the most comprehensive information to support the decision-making process. Data is a valuable asset for stakeholders, either to valorise them in scientific papers - which are important for recognition and promotion within the scientific community - or for showcasing their work to attract funding from international aid. Consequently, "*there is this perception that each sector tends to keep its information* [surveillance data] *for itself, and that's a brake to the operationalization of the AMR surveillance system*" (Respondent 8).

Regarding data sharing, interview contents suggested that there was a need to better define the roles and responsibilities of the various stakeholders in the national One Health surveillance system, as mentioned by Respondent 24: "*It will be sufficient to have a regulatory framework that lays out what data is to be collected, how it will be collected, and which institutions are to be involved in the management of data.*"

## Discussion

By combining a review of legal documents and interviews with stakeholders, this study provides an overview of the political-institutional framework for the governance of ABR in Senegal, and of its capacity to manage ABR using a One Health approach. Since 2017, Senegal has been committed to an integrated, cross-sectoral approach to ABR. Although this is only partially functional, Senegal already has multi-sectoral institutional mechanisms at central and local levels to guide and monitor ABR-related policy implementation. An updated national policy to combat ABR exists, but its operational implementation remains partial. The regulatory framework is comprehensive, covering the areas of prevention, use and monitoring, but is not necessarily always adapted and appropriately enforced. As ABR is a major international concern, the country receives considerable technical and financial support from external partners, particularly international organizations such as the FAO and WHO and development agencies such as USAID, to develop and implement ABR-related policies. This external influence can compromise its abilities to develop policy solutions aligned on national priorities and a shared vision to address the ABR issue.

The evaluation of Senegal's capacities to manage ABR from a One Health perspective is based on the perception and posture of the different categories of actors involved in ABR governance in the country. However, some major actors in the human health sector declined to participate in the study. Furthermore, we cannot certify that the discourse of those who did participate was a typical representation of the views of their institution.

This study highlighted the central position of the national One Health Platform in the governance landscape of ABR in Senegal. This type of multi-sectoral institutional mechanism has been established in many other African countries with the strong support of the TFPs, in particular the Tripartite [The Tripartite consisted of the first three members of the current Quadripartite mentioned on page 4, namely the WHO, the Food and Agriculture Organization (FAO), and the World

Organisation for Animal Health (WOAH), from 2010 to 2022, after which the United Nations Environment Programme (UNEP) joined as a fourth member.] with the financial support of USAID and the World Bank through the REDISSE (Regional Disease Surveillance Systems Enhancement) project [39–41]. The establishment of such a mechanism is also a means for countries to commit to global policies, such as the Global Health Security Agenda, and to comply with the implementation of the IHR [42]. All of these national platforms have more or less the same organizational structure and share a number of difficulties, which vary in severity from country to country. As in Senegal, they usually include a permanent secretariat and thematic groups modelled on the IHR's technical areas, including one dedicated to AMR. However, they differ in their governance arrangements. In some countries, including Senegal, they are headquartered at the highest level of government, in the Prime Minister's Office. In others, they are hosted within a ministry (usually the one responsible for public health, as in Burkina Faso or Kenya), with a more-or-less rotating secretariat among sectors, as is the case in Uganda [43–45]. Some countries have begun to decentralize these mechanisms to the local level (e.g., Kenya and Senegal). As underlined in this study and in others, these platforms have undeniably improved the governance of complex health hazards, by providing a forum for discussion and exchange for all stakeholders, including governmental authorities, professionals, technical and scientific institutes and the civil society [44,45]. But they are also facing numerous difficulties, such as: (i) a lack of domestic resources, which makes them highly dependent on external resources and compromises their autonomy and sustainability [46]; (ii) a lack of legal framework clearly defining their roles and missions and of the political support, which brings confusion regarding the platform's leadership for orienting and implementing One Health interventions; (iii) a lack of a strategic vision taking into account concerns and priorities of the whole array of stakeholders, including actors that are not traditionally considered to be at the centre of global health issues [47]; and (iv) a lack of specific capacities in managing multi-stakeholder platforms, in policy development and evaluation, and in facilitating science-society-policy dialogue [48]. In addition, in some countries, the tasks assigned to the platforms have come to duplicate the national epidemic management committees already in place, resulting in the dilution of already limited resources [49]. As a result, these platforms have not succeeded in fully fulfilling the role they were assigned in terms of governance of complex health issues, including ABR.

The most comprehensive legal instrument developed in Senegal to manage ABR with a One Health perspective is the National Action Plan to combat AMR issued in 2020, following the adoption of the Global Action Plan by the WHO. However, the implementation of this plan has been considerably slowed by a lack of domestic funding, the reallocation of budget to the management of the COVID-19 crisis, a weak appropriation of the measures at local level, and a lack of alignment with initiatives developed locally [50]. This difficulty in implementing national policies, directly translated from global policy, is not unique to Senegal and has been described in many other resource-limited African countries [51–53].

The results of this study suggest that a strengthened institutional framework appears essential for effective ABR governance. Participants emphasized the need for an inter-sectoral institutional mechanism backed up by political support and a legal framework specifying the role and missions of all stakeholders, a revised and enforced regulatory arsenal, and more effective health systems. They also expressed the need for more inclusive and equitable participation across sectors, professions, and decision-making levels in policy formulation for the management of ABR. This is in line with the strengthening of One Health institutionalization advocated by the scientific community and development actors [26,54–56]. However, institutionalization must be properly managed to produce the desired effects and not to cause adverse effects.

First, institutionalization must come with sufficient additional domestic resources. In Senegal, human and financial resources in technical departments of ministries and in other organizations involved in ABR One Health governance are drastically insufficient to allow engagement of the staff in intersectoral activities without compromising the good implementation of their primary missions. This is exacerbated by the fact that stakeholders are overburdened with a multiplicity of ABR-related activities proposed by different TFPs in a context where international aid is still too often fragmented, uncoordinated and even in competition [54,57,58] and incentivizes the participation of national counterparts [59]. In general, LMICs rely mainly on external funding to develop, implement and evaluate ABR-related policies, and this makes them

vulnerable in terms of political sovereignty as they have to shape their politico-institutional framework in light of the funding available [60]. Additionally, as this funding is not sustainable over time because it is backed by short-term programmes, the politico-institutional framework may be disrupted when these programmes come to an end, because domestic funding is unable to take over [43,44,60,61]. In Senegal, the One Health platform's annual work programmes and budget for 2024 include 189 activities, of which only a small proportion is either funded (2%) or co-funded by the state (17%). In this context of scarce resources, the institutionalization of One Health governance can generate competition between actors working in the technical directorates of the ministries and the national One Health platform. Additionally, part of the technical and financial aid that used to go directly to sectoral ministries is now channelled through the national One Health platforms. This could potentially create frustration among these players in the long term and lead them to refocus on their sector-specific activities.

Secondly, institutionalization must enable the establishment of rules for collaboration and accountability, while providing a flexible and non-restrictive framework. Although binding legal instruments framing the roles and responsibilities of actors can improve their participation in certain collective activities, this does not ensure their commitment to working collectively to find and implement political solutions to the ABR issue [61]. Collaboration may feel more of an obligation than an opportunity to share points of view and expertise in the service of collective action. Additionally, people are more reluctant to collaborate when the rules are laid down in a binding document, because they feel they are trapped in interactions from which they cannot withdraw [54].

Finally, One Health institutionalization for ABR management inevitably requires the development of global policies to which countries, whatever their socio-economic context, would have to adhere [62]. For ABR, where the different contexts are even more varied than for other health issues, the difficulty will lie in developing policies that are equitable across countries and populations and that take into account the diversity of socio-economic and political contexts [26]. The historical prioritization of the needs of high-income countries over those of LMICs in global policies risks leading to the development of ABR-related policies that focus on the benefits for a restricted fraction of the world, such as policies focusing on the conservation of critical antibiotics at the expense of improving access to antibiotics and mortality [59,63], and that can be implemented only in contexts that show comprehensive legal frameworks, adequate enforcement capacities, low corruption levels, and strong healthcare systems [64]. However, our study underlined that this pre-requisite context was missing in Senegal, as in many other LMICs. Consequently, in a bid to comply with international standards, or influenced by TFP-organized workshops, countries with limited resources can find themselves compelled to adopt ABR-related policies with unattainable objectives that are ill-suited to their implementation capacity or to their needs and priorities [65]. Additionally, as TFPs align their support with global policies, there is always a risk that a given government will be deprived of funding if it does not align itself with the priorities of international funders [59]. Even when not binding, these global policies and strategies may have a strong influence on national policy frameworks. Indeed, they are often accompanied by evaluation tools, such as the JEE that assesses the country compliance with the IHR and the WOAH's norms, or the FAO Progressive Management Pathway for AMR (FAO-PMP-AMR) that monitors the implementation of the FAO Action Plan on AMR 2021–2025 at country level. The evaluations carried out in the countries using these tools to check their level of compliance with the related policies lead to evidence and recommendations that are then used by TFPs to plan their future support to the country, or by the countries themselves to propose interventions that are more likely to be attractive for donors. This leaves little room for manoeuvre for countries to develop policy solutions and interventions more tailored to the local context, for instance by engaging with populations affected by ABR or with authorities operating at the most local level over policy design [56,60]. In Senegal, the selection of measures to be included in the second national action plan to combat AMR was mainly driven by the availability of external fundings for this type of measure, with a limited participation of local actors. Additionally, these evaluation tools also place countries in a race for ratings, sometimes to the detriment of other national priorities [66]. Indeed, good scores make them more attractive than countries with lower scores for the implementation of programmes to support the fight against ABR [57,58].

Consequently, the reinforcement of institutionalization of One Health governance must be accompanied by particular attention to participation, collaboration and coordination, accountability and transparency, sustainability, and equity [55] and must avoid strengthening the powers of global actors, with the creation of competitive spaces at the international and national levels [59]. The development of policy solutions must better articulate the visions and priorities of actors at the different decision-making levels, as well as those of the community. To this end, One Health platforms must position themselves as frameworks for consultation that allow all parties to freely raise their voices in order to define a concerted vision for an effective management of ABR at country level. At the same time, donors must give countries more latitude to develop bottom-up policy solutions rooted in local realities and require central administrations to involve contextual experts in policy formulation workshops. While evaluation is a necessary step in the policy cycle to provide evidence-based information and to guide policy development and implementation, new frameworks and tools are needed to undertake comprehensive political and socio-ecosystemic evaluation and guide the development of effective One Health policies [67].

This study has certain limitations.

Firstly, certain categories of actors were underrepresented or not represented at all in our interviews. Several institutions in the human health sector did not wish to participate in the study. Health professionals working at the bottom of the health pyramid, as well as local authorities and civil society, were not included. We therefore did not capture all perspectives on the topic under study, and greater involvement of local stakeholders would have provided a different perspective on the operationalization of the One Health concept in the context of ABR management.

Secondly, certain topics proved sensitive for respondents, particularly when it came to expressing their views on the capabilities of the One Health platform or the influence of TFPs on national policies. In this context, despite the fact that the confidentiality of their comments was assured, it is possible that they did not fully express their opinions.

It is also important to note that One Health governance in Senegal is a highly evolving policy mechanism due to various internal and external factors that can influence its dynamics, such as political changes, interpersonal relationships, health crises, aid programmes, and new international or national policy instruments. The results of this study are therefore only valid for the period during which the data was collected.

## Conclusions

In an international context calling for One Health governance to combat ABR, Senegal has established a politico-institutional framework aligned with international standards. However, the following can be considered critical to effectively manage ABR from a One Health perspective: enhancing the capacity and the autonomy of the national One Health platform; developing more inclusive and equitable policy solutions; strengthening the regulatory framework and its enforcement; and improving healthcare infrastructure.

This suggested that institutionalization was foundational to suport effective One Health governance of ABR. However, institutionalization is not without risk. At a global level, strengthening global policies and their related evaluation tools can have an unintended negative impact on the relevance and sustainability of the politico-institutional frameworks within countries. At a national level, it can lead to competition and a dispersal of resources between actors, which is detrimental to the collective action that underpins One Health governance. A strengthening of the institutionalization of One Health governance for ABR must be accompanied by a reform of the process of developing and implementing global and national policies for a better representation of interests and priorities.

## Supporting information

**S1 Table. Regional and national institutions involved in antibiotic resistance management and their main legal instruments.**
(DOCX)

**S2 Table. Surveillance programmes for antibiotic resistance, use, quality and residues in Senegal.**
(DOCX)

## Acknowledgments

The authors would like to thank all the participants, and Kadiatou Barry who helped with the transcriptions of the interviews.

## Author contributions

**Conceptualization:** Mouhamadou Moustapha Sow, Marion Bordier.

**Data curation:** Mouhamadou Moustapha Sow.

**Formal analysis:** Mouhamadou Moustapha Sow, Marion Bordier.

**Funding acquisition:** Mamadou Ciss, Marion Bordier.

**Investigation:** Mouhamadou Moustapha Sow, Léo Delpy, Mamadou Ciss, Marion Bordier.

**Methodology:** Mouhamadou Moustapha Sow, Léo Delpy, Marion Bordier.

**Project administration:** Mamadou Ciss, Marion Bordier.

**Software:** Mouhamadou Moustapha Sow, Marion Bordier.

**Supervision:** Assane Gueye Fall, Nicolas Djighnoum Diouf, Ndeye Mery Dia, Marion Bordier.

**Validation:** Mouhamadou Moustapha Sow, Léo Delpy, Marion Bordier.

**Visualization:** Mouhamadou Moustapha Sow, Marion Bordier.

**Writing – original draft:** Mouhamadou Moustapha Sow, Léo Delpy, Jean-Hugues Caffin, Marion Bordier.

**Writing – review & editing:** Mamadou Ciss, Assane Gueye Fall, Nicolas Djighnoum Diouf, Jean-Hugues Caffin, Ndeye Mery Dia, Marion Bordier.

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
