## [Decision Letter · Decision Letter 0]

25 Nov 2025

PGPH-D-25-03107

One Health governance of antibiotic resistance in Senegal

Dear Dr. Bordier,

Thank you for submitting your manuscript to PLOS Global Public Health. After careful consideration, we feel that it has merit but does not fully meet PLOS Global Public Health’s publication criteria as it currently stands. Therefore, we invite you to submit a revised version of the manuscript that addresses the points raised during the review process.

Please carefully review all comments provided by the Editor and Reviewers (see below) and address them in revision, providing adequate justification in your rebuttal letter.

We look forward to receiving your revised manuscript.

Kind regards,

Giorgia Sulis, M.D., Ph.D.

Academic Editor

Journal Requirements:

i. Please clarify all sources of financial support for your study. List the grants, grant numbers, and organizations that funded your study, including funding received from your institution. Please note that suppliers of material support, including research materials, should be recognized in the Acknowledgements section rather than in the Financial Disclosure.

ii. State the initials, alongside each funding source, of each author to receive each grant. For example: "This work was supported by the National Institutes of Health (####### to AM; ###### to CJ) and the National Science Foundation (###### to AM)."

iii. State what role the funders took in the study. If the funders had no role in your study, please state: “The funders had no role in study design, data collection and analysis, decision to publish, or preparation of the manuscript.”

iv. If any authors received a salary from any of your funders, please state which authors and which funders.

2. Please ensure that your Ethics Statement is available in its entirety at the beginning of your Methods section, under a subheading 'Ethics Statement'.

3. In the online submission form, you indicated that “The full dataset generated during the current study is available from the corresponding author on reasonable request.”.

3. Uploaded as supplementary information.

Additional Editor Comments (if provided):

This is an interesting manuscript addressing a very important topic. However, there are aspects of the manuscript that require revision to enhance clarity and contextualization.

All three reviewers have highlighted issues in the discussion section that I completely agree with. Please review their comments and address them carefully. Other areas that require clarifications are related to the methods used, as more details should be provided for greater transparency and for a better interpretation of the study findings.

In preparing the revised manuscript, I encourage you to review the instructions for authors and ensure compliance with those requirements. Please prioritize these whenever the reviewers request formatting changes that are in disagreement with the journal's requirements. For instance, while one reviewer recommends to revise the abstract to make it structured, our journal indicates that "While the Abstract is conceptually divided into three sections (Background, Methodology/Principal Findings, and Conclusions/Significance), do not apply these distinct headings to the Abstract within the article file.". You can find more details here: https://journals.plos.org/globalpublichealth/s/submission-guidelines.

I also invite you to review the requirements for key disclosures. Please revise the financial disclosure to include all required information: this statement must describe the sources of funding for the work included in this submission and the

role the funder(s) played, including grants and any commercial funding of the work or authors.

Regarding data availability, please note that Authors are required to make fully available and without restriction all data underlying their findings. Please see our PLOS Data Policy page for detailed information on this policy. A Data Availability Statement, detailing where the data can be accessed, is required and needs revision.

Reviewers' comments:

Reviewer's Responses to Questions

**Comments to the Author**

1. Does this manuscript meet PLOS Global Public Health’s publication criteria?

Reviewer #1: Yes

Reviewer #2: Yes

Reviewer #3: Yes



Reviewer #1: Yes

Reviewer #2: N/A

Reviewer #3: Yes

3. Have the authors made all data underlying the findings in their manuscript fully available (please refer to the Data Availability Statement at the start of the manuscript PDF file)?

Reviewer #1: Yes

Reviewer #2: Yes

Reviewer #3: Yes

4. Is the manuscript presented in an intelligible fashion and written in standard English?

Reviewer #1: Yes

Reviewer #2: Yes

Reviewer #3: Yes

5. Review Comments to the Author

Reviewer #1: This is a well-written manuscript overall. It clearly describes the knowledge gap with AMR governance in Senegal. The methods are detailed and the results are clear. There are a few comments:

- Line 242: Is it AMR stewardship of an antimicrobial stewardship plan?

- The discussion section (461-492) is not clear. It does not speak to Senegal but to the African continent as a whole. The section should tie the findings in this paper to the continental ecosystem. The same is noted from lines 493 - 551; it focuses more on the international ecosystem. The discussion should focus on what these findings mean and then flow into the recommendations

- The recommendations are not clearly stated, and the conclusion still highlights the global level. Except this can be linked to the situation in Senegal, then it is not clear on how these recommendations impact the local situation

Reviewer #2: Overall: Very detailed and insightful article about AMR governance in Senegal. The article uses a grey literature search and stakeholder interviews which add to its strengths. Major feedback includes better structuring of the discussion that interpret the studies findings more clearly. Would suggest attempting to shorten the article if possible and restructuring some long run-on sentences. I would also move some of the claims made in the results to the discussion.

Abstract:

1. Would be helpful to have a structured abstract if possible.

Introduction:

1. Would break the sentence in lines 66 - 69 into two sentences: "One Health is defined as an integrated, unifying approach that aims to sustainably balance and optimize the health of people, animals, and ecosystems. The One Health concept recognizes the links between and interdependence of the health of humans, domestic and wild animals, plants, and the wider environment (including ecosystems)."

2. Would do the same with lines 70 - 75 as above. Currently the sentences is very long.

2. Line 81 should start with a new paragraph about what Senegal is doing.

Methods:

1.With the methods, you begin with defining the concepts but I would start with your actual methodology (lines 107 to 112). This paragraph should stand alone and the definitions can come after.

2. Lines 97 & 99 would suggest just calling it ABR instead of ABR issue

3. Line 101 would remove "and" and place a comma: "sometimes underpinned by an institutional and legal framework, and facilitated by effective management, transparent mechanisms, and relationships."

3. Its seems like this is more of a grey lit review and not an academic lit review, I might specify this.

4. Was academic literature assessed at all? And if not, why?

5. Would recommend creating a flow diagram showing the number of experts invited, number who participated, number that conducted in-person interview, and number who did not. In this diagram I would also include expert counts with the sector and occupation the expert belongs to so readers can visualize this data.

6. For interviews not conducted in person, would add a line on how they were carried out (e.g. phone, virtual platform, written).

7. Remove lines 159 - 163 as you have and ethics section and this is repeated.

Results:

1. The first header "The politico-institutional framework for antibiotic resistance management in Senegal" aligns with the first objective in the methodology but the subsequent headers don't. I would keep the headings consistent with your objectives and add subheadings as needed to improve clarity (i.e. second header in results: Drivers of One Health governance and ABR management, third header: Improving effective One Health governance).

2. Line 329 sounds close to a claim made in a discussion section not the results. Would remove and start with the next line 'However, most programs..." similar for line 349, would recommend removing and presenting the finds in lines 352. If these statements are from your interviewees, then I would directly specify that and present it as such, again it reads like author comments which should be held for the discussion.

3. Lines 377 - 385, 394 - 397, 424 - 426 similarly to #2, read more like a discussion section or like a recommendation that would be brought forward in the discussion section rather then the results section. I would recommend beginning the section with the objective findings of your Key Stakeholder interviews.

4. Lines 377 - 385 need references unless all of these claims were from the interviews -if so this needs to be stated. Right now, line 388 discusses hospital laboratories but I can't find any interview excerpts that speak to health system strengthening from COVID or Ebola. So the lines read like the author is making these claims. If these are the authors claims, would suggest moving these down to the discussion and adding citations here.

Discussion:

1. There is a lot of material here but the discussion section doesn't explicitly state the key findings. This section very rarely engages with the data generated through the interviews and I can't see where it mentions the 5 key themes that were introduced in the results. The section reads largely as a theoretical or literature-based commentary. To strengthen it I would recommend explicitly interpreting each key finding of your results (here is where you could add some commentary that was in the results section), integrating what interviewees said, and highlighting the study’s contributions to AMR governance.

2. I don't think you need subheadings in the discussion section and if keeping, I would again match them up to your original objectives.

3. Line 489 needs a citation.

4. Some language in this section should be more normative to maintain an analytical tone (e.g. "This calls into question the legitimacy..." "....rather than real strategic governance instruments"), all true but would try to keep more neutral language.

5. Line 545 is great, it speaks to the barriers countries face when tailoring their policies to the local context. I would build on that, reiterate what this study adds to the current literature, and what future work is needed beyond this study to continue to build and improve AMR governance within Senegal and other LMICs.

6. Lines 548, I agree that chasing a score is not the goal but might a public score increase transparency, which is needed in AMR governance and policy? I might mention that here.

7. Need a citation and maybe example for this claim in line 549: "...sometimes to the detriment of other national priorities.."

8. A limitation section is needed.

Conclusion:

1. Would try to shorten this to one concise paragraph.

Reviewer #3: This is a well-written and well-designed study - in the returned PDF attachment I have highlighted some minor corrections that are required to be made prior to acceptance for publication. Beyond these minor typographical and grammatical edits, there are two minor changes that I would like to see made before the manuscript can proceed to publication.

The first, is just to add some further clarifications around the methodology employed. As highlighted in the returned PDF this is regards to the number of interviews and interviewees, giving the reader a clearer indication of how many interviews were conducted in a group setting and subsequently some clearer outline of what languages the interviews were conducted in, and if not carried out in English, how and when the translation process took place (pre-transcription, pre-analysis, or just for the purpose of reporting quotes in the manuscript)?

The second, not outlined in the returned PDF, is that wherever possible/relevant in the Discussion section could you add more content on the main findings of the interviews. Better integrating the key themes from the interviews into the sections primarily focused on the policy analysis and implementation would further bolster the arguments made and make the connection between the two main methodological components of the research clearer for the reader.

Once these relatively minor changes have been made, I would be happy for the paper to proceed to publication.

6. PLOS authors have the option to publish the peer review history of their article (what does this mean? ). If published, this will include your full peer review and any attached files.

**Do you want your identity to be public for this peer review?** For information about this choice, including consent withdrawal, please see our Privacy Policy .

Reviewer #1: **Yes:** Abiodun Egwuenu

Reviewer #2: No

Reviewer #3: No

Figure Resubmissions:

---

## [Decision Letter · Decision Letter 1]

18 Feb 2026

Governing Antibiotic Resistance Through One Health: Insights from the Political and Legal Landscape in Senegal

PGPH-D-25-03107R1

Dear Dr Bordier,

We are pleased to inform you that your manuscript 'Governing Antibiotic Resistance Through One Health: Insights from the Political and Legal Landscape in Senegal' has been provisionally accepted for publication in PLOS Global Public Health.

Best regards,

Giorgia Sulis, M.D., Ph.D.

Academic Editor

Thorough copy-editing for grammar and style errors is strongly recommended when article proofs become available. See advice for Reviewers in this regard.

Reviewer Comments (if any, and for reference):

Reviewer's Responses to Questions

**Comments to the Author**

1. If the authors have adequately addressed your comments raised in a previous round of review and you feel that this manuscript is now acceptable for publication, you may indicate that here to bypass the “Comments to the Author” section, enter your conflict of interest statement in the “Confidential to Editor” section, and submit your "Accept" recommendation.

Reviewer #1: All comments have been addressed

Reviewer #2: All comments have been addressed

Reviewer #3: All comments have been addressed

2. Does this manuscript meet PLOS Global Public Health’s publication criteria ? Is the manuscript technically sound, and do the data support the conclusions? The manuscript must describe methodologically and ethically rigorous research with conclusions that are appropriately drawn based on the data presented.

Reviewer #1: Yes

Reviewer #2: Yes

Reviewer #3: Yes

3. Has the statistical analysis been performed appropriately and rigorously?

Reviewer #1: Yes

Reviewer #2: N/A

Reviewer #3: Yes

4. Have the authors made all data underlying the findings in their manuscript fully available (please refer to the Data Availability Statement at the start of the manuscript PDF file)?

Reviewer #1: Yes

Reviewer #2: Yes

Reviewer #3: Yes

5. Is the manuscript presented in an intelligible fashion and written in standard English?

Reviewer #1: Yes

Reviewer #2: Yes

Reviewer #3: Yes

6. Review Comments to the Author

Reviewer #1: Thank you for addressing all the comments that were raised and for collating insights from the politico-legal perspective, which is often neglected.

Reviewer #2: The publication meets PLOS Global Public Health's criteria for publication. The authors have reported where the data is publicly available and the manuscript has been edited well and reads nicely. Thank you for addressing my comments and suggestions.

Reviewer #3: Thank you for addressing my comments in a collegial and thorough manner. I still disagree with your assertion that one interview represents one institution, and that where multiple people were present that they can be considered one corporate perspective. One of the major strengths of the semi-structured interview methodology is that it can help us narrate divergence between communicated institutional practices/expectations and actual practices on the ground. However, this difference of opinion is likely due to disciplinary differences/norms and should not delay the publication of this article further.

7. PLOS authors have the option to publish the peer review history of their article (what does this mean? ). If published, this will include your full peer review and any attached files.

**Do you want your identity to be public for this peer review?** For information about this choice, including consent withdrawal, please see our Privacy Policy

Reviewer #1: **Yes:** Abiodun Egwuenu

Reviewer #2: No

Reviewer #3: No
